# Incidence of straight and angulated screw channel following immediate implant placement in esthetic zone: A simulated cone-beam computed tomography study

**Watcharapon Radomsutthisarn**[1], **Kornkamol Kretapirom**[2], **Pranai Nakaparksin**[1] *

1 Department of Advanced General Dentistry, Faculty of Dentistry, Mahidol University, Bangkok, Thailand,
2 Department of Oral and Maxillofacial Radiology, Faculty of Dentistry, Mahidol University, Bangkok, Thailand

☯ These authors contributed equally to this work.
* Pranai.nak@mahidol.ac.th

**Data Availability Statement:** All relevant data are within the manuscript and its Supporting Information files.

## Abstract

### Statement of problem

The demand for immediate implant placement (IIP) in the esthetic zone is rapidly increasing. Despite the treatment's benefits, the sagittal root position often dictates implant angulation, commonly necessitating the use of cement-retained restorations. This study investigates the impact of angulated screw channel on IIP in the esthetic zone.

### Purpose

The purpose of this cone-beam computed tomography (CBCT) study was to determine the probability of IIP in the esthetic zone, using four different implant angulations.

### Materials and methods

A retrospective review of CBCT images was conducted and accessed on 9 June 2021. The midsagittal images of maxillary anterior teeth were input into an implant planning software (Implant Studio®). Bone Level X Straumann® implant (BLX 3.5, 3.75, 4.0, 4.5, 5.0 mm) and Bone Level Tapered implant (BLT 2.9, 3.3 mm) were selected for 3D implant planning of IIP in the esthetic zone. The frequency distribution and probability of IIP were recorded and compared among all maxillary anterior teeth.

### Results

CBCT images from 720 teeth (120 patient) were evaluated, revealing an overall probability of IIP in the esthetic zone is 76.11% (548/720). Different implant restoration type was evaluated in this study, with the following results; straight screw-retained prosthesis at 3.47% (19/548), cement-retained prosthesis at 14.59% (80/548) and angle screw-retained prosthesis at 85.40% (468/548).

**Funding:** The authors received no specific funding for this work.

**Competing interests:** All authors including, Dr. Pranai Nakaparksin, Dr. Watcharapon Radomsutthisarn, Dr. Kornkamol Kretapirom have declared that no competing interests exist.

## Conclusion

IIP with traditional straight screw-retained prosthesis demonstrated the lowest probability. Nevertheless, the use of angulated screw channels enhances the probability of achieving straight screw-retained prostheses.

## Clinical implications

The angulated screw channel is essential for increasing the probability of screw-retained prosthesis in IIP in the esthetic zone. However, limitation in screw angle correction still necessitate the use of cement-retained prostheses for numbers of patients undergoing IIP.

## Introduction

Immediate implant placement (IIP) and immediate provisionalization following tooth extraction has become a common clinical procedure for the replacement of failing teeth with high survival rate [1–4]. The most recent consensus conference in 2023 has recommended the protocol as the treatment of choice when the ideal conditions are present [4]. However, placement of dental implants in the maxillary anterior esthetic zone is challenging for clinicians, due to the patients' esthetic expectations and several risk factors that affect overall outcome of the treatment. Therefore, for desired esthetic results and tooth function when performing an IIP [5, 6], this procedure requires strict patient and site-specific selection criteria [3].

To achieve successful tissue integration and minimize risk of complications for IIP, site-specific criteria should be followed, including absence of gingival recession, adequate gingival phenotype, sufficient bone anchorage, presence of facial bone wall, and absence of infection [3]. However, the complexity and technique of the surgical procedure can be challenging. Subsequently, an appropriate implant diameter and length selection has shown to affect both esthetic consequence and primary stability [3, 7, 8]. Smaller diameter implant is recommended for adequate grafting material between facial bone and planned implant position [9]. For predictable implantation in IIP, 4–5 mm implant engagement at apical bone beyond tooth socket is recommended [3].

A preoperative CBCT scan provides valuable information for implant planning including implant position, diameter, length, and angulation [10, 11]. Implant angulation is dictated by sagittal root position (SRP), in turn results in either cement-retained or screw-retained implant prosthesis [12]. Despite the lack of evidence from systematic reviews indicating a difference in peri-implant marginal bone loss between cement-retained and screw-retained implant crowns [13–18], a positive correlation was found between excess subgingival residual cement and peri-implant disease. Thus, clinician commonly plan for implant angulation to accommodate for screw-retained implant prosthesis.

A recent 2-dimentional digital implant template study has demonstrated low possibility of straight screw channel screw-retained restoration for IIP protocol in the esthetic zone [19]. A follow up study using angulated screw channel has shown to increase the frequency of screw-retained implant prosthesis [20].

The purpose of this study was to determine the probability of IIP in the maxillary anterior teeth, using four different implant angulations, traditional screw-retained implant prosthesis (screw-retained), traditional cement-retained implant prosthesis (incisal edge), limited angulation for angulated screw channel (angle screw), and limit restorability angulation for cement-retained implant prosthesis (middle half).

## Material and method

### Patient selection

This retrospective study was approved by the Institutional Review Board (IRB) of the Faculty of Dentistry/Faculty of Pharmacy, Mahidol University (IRB 2021/052.0906). The sample for this study were patients (39 males and 81 females) with a mean age of 35 years old (range: 21 to 77 years old) who had CBCT scans of the anterior maxilla at the Oral and Maxillofacial Radiology clinic, Faculty of Dentistry, Mahidol University performed during January 2012 – August 2020. CBCT images were performed by 3D Accuitomo 170 (J. Morita®, Japan) and selected for evaluation on 9 June 2021. The design of this study has not been performed. So, there will be no similarity in statistical data for this study. The retrospective study of Asok Velayudhan et al [21], which the objective related to this study have been used for calculated sample size. The sample size calculation for estimating proportion used as shown in the formula:

$$n = \frac{z_{1-\frac{\alpha}{2}}^2 p(1-p)}{d^2}$$

| | | |
|---|---|---|
| $\alpha$ | : | Confident interval = 0.05 |
| $Z_{1-\alpha/2}^2$ | : | Standard Normal score at p = $1 - \alpha/2$ |
| p | : | Proportion from previous study |
| d | : | margin of error |

The output of sample size calculation is 114. The 120 CBCT scans are sufficient for this study. Inclusion criteria were as follows:

- CBCT must have field of view (FOV) 6 cm. x 6 cm. (voxel size 0.125 mm) or FOV 8 cm. x 8 cm. (voxel size 0.160 mm.).

- The patient must be at least 18 years old at the time of CBCT scan.

- The patient must have all maxillary anterior teeth (canine to canine) with stable posterior occlusal support.

- The patient must have a normal alignment of maxillary anterior teeth without periodontal or periapical pathologies as observed on CBCT.

Exclusion criteria were as follows:

- Radiographic evidence of periapical inflammatory lesion, periodontally involved teeth, severe root resorption, and/or previous records of trauma, or bone pathology.

- Radiographic evidence of artifacts and orthodontic appliances.

- Radiographic evidence of surgical (guided bone/tissue regeneration) treatment in the maxillary anterior dentition.

- Upper anterior teeth were misaligned.

- CBCT images that were distorted or blurred.

### Data reconstruction from and CBCT images and digital simulation

CBCT volumetric data was exported to DICOM files and reconstructed by using Implant Studio® (3Shape®, Denmark). For the axial view, the implant was positioned at the center of the

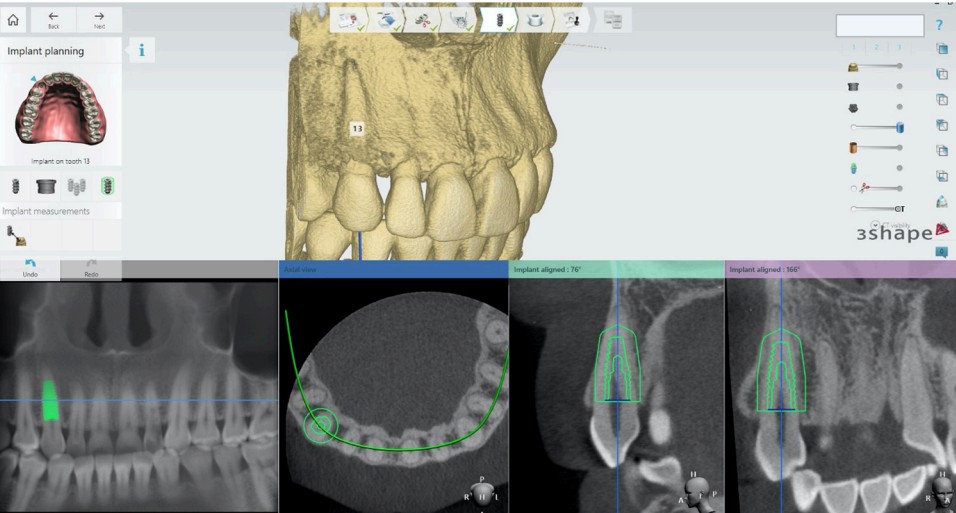

**Fig 1. User interface of Implant Studio® software showing the implant placement at area 13.** In the axial view, the implant was positioned at the center of the selected tooth. In the panoramic view, the implant was aligned parallel to the long axis of tooth. In the coronal view, the implant was rotated parallel to the long axis of the tooth. The mid-sagittal image, which was been sectioned at the bucco-lingual aspect of the tooth, was used to measure and plan of IIP.

selected tooth. In the panoramic view, the implant was aligned parallel to the long axis of the tooth. In the coronal view, the implant was rotated parallel to the long axis of the tooth. The mid-sagittal image, which was located at the bucco-lingual side of the tooth, was used to measure all parameters (Fig 1).

The mid-sagittal of the selected tooth was used for the measurement of the buccal bone thickness and faciopalatal root dimension at the bone crest level. SRP was classified with respect to the alveolar housing according to Kan et al. [12].

## Implant planning and determining of implant diameter and length

Implant was planned by Implant Studio® (3Shape®, Denmark) using Bone level X Straumann® implant (BLX 3.5, 3.75, 4.0, 4.5, 5.0 mm) (Straumann®, Switzerland) and Bone Level Tapered implant (BLT 2.9, 3.3 mm) (Straumann®, Switzerland). BLT 2.9 mm was used in lateral incisor only. The angulated screw channel was included in the calculation of all implant diameter except diameter of 2.9 mm was not included.

An implant should be placed 3 mm apical and 2 mm palatal to gingival margin [22]. Thus, the 3 ± 0.02 mm line apical to gingival margin and a 2 ± 0.02 mm line, which perpendicular to the first line were created. The end point was marked as a position of the most buccal aspect of implant platform point P. The line drawn from incisal edge to root apex was long axis of the tooth. The point between incisal edge to CEJ was marked as point M. The point between the center of the most prominent point of cingulum to the incisal edge was marked as point S. The implant component points and lines are shown in Fig 2.

The implant was positioned in four different angulations (Fig 3) as follows: screw-retained, incisal edge, angle screw and middle half. In traditional cement-retained implant prosthesis (incisal edge), long axis of implant will coincide with incisal edge. In traditional screw-retained implant prosthesis (screw-retained), long axis of implant will coincide with point S. In limit angulation for angulated screw channel (angle screw), long axis of implant will coincide at the point that an angle up to 25° from traditional screw-retained. In limit restorability angulation

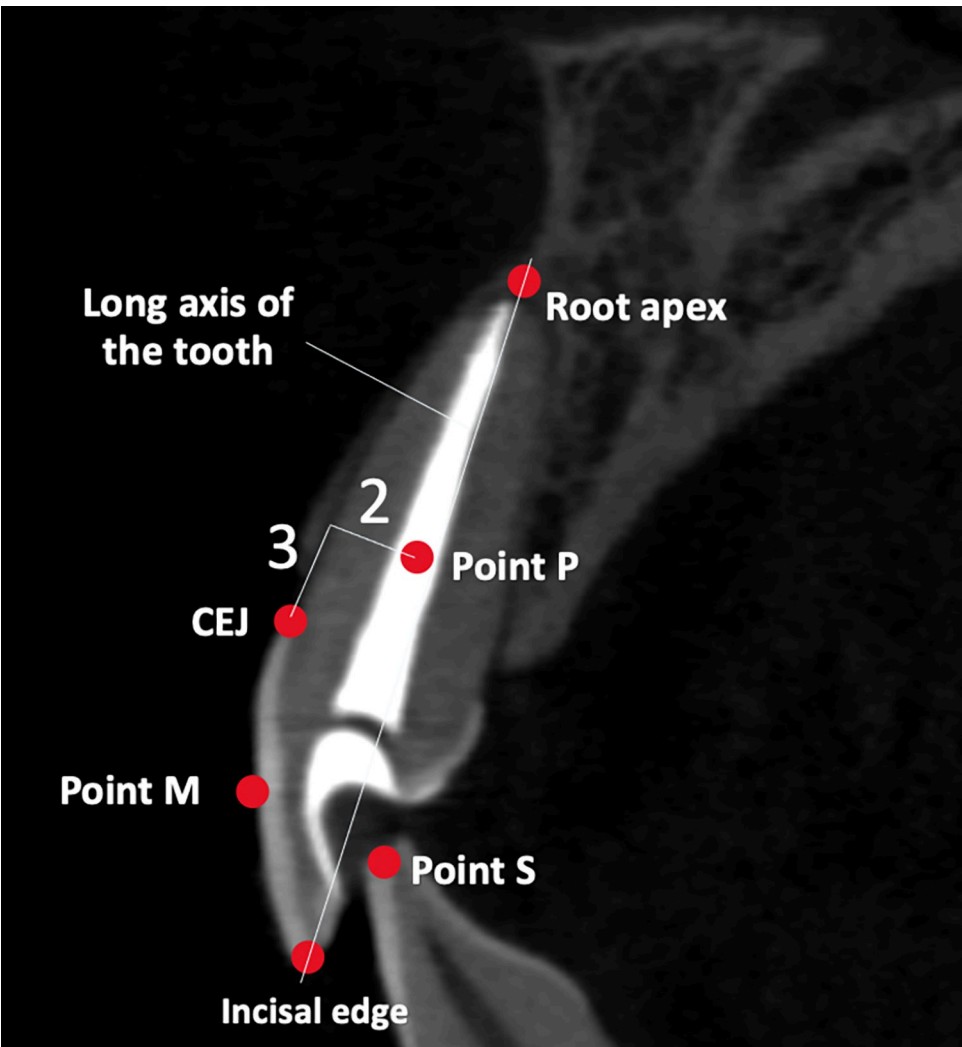

**Fig 2. The component points of the implant are marked in the mid-sagittal images.**

for cement-retained implant prosthesis (middle half), long axis of implant will coincide with point M.

To determine the implant diameter and length, the smallest implant diameter and length were selected. The implant length was increased until engaging a minimum of 4 ± 0.02 mm in distance from the apical to root apex [3], which is considered as the first implant position and increased until the apex of implant is positioned 1 mm away from critical structures. After that, increased the next size implant diameter until it reached the maximum, which calculated by faciopalatal root dimension minus two to ensure sufficient bone grafting gap [3].

All measurements (the buccal bone thickness and faciopalatal root dimension) for the implant planning process were performed by one examiner. The intraexaminer reliability was assessed by repeating the measurement on 10 randomly selected CBCT images 1 month after the first measurement by intra-class correlation coefficient (ICC).

Descriptive statistics were used to report the probability of IIP in the esthetic zone using four different implant angulations as percentages.

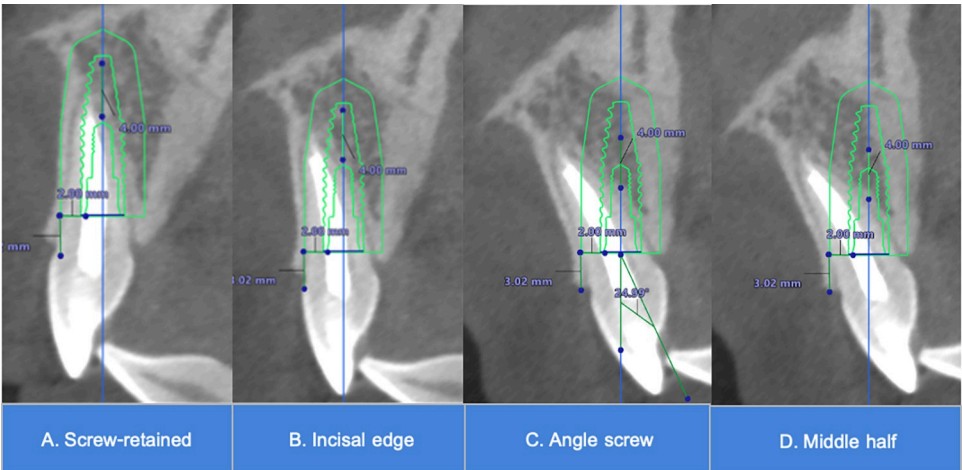

**Fig 3. The four different implant angulations in this study.** A. Screw-retained, B. Incisal edge, C. Angle screw, D. Middle half.

## Results

120 CBCT images from maxillary anterior teeth (240 central incisors, 240 lateral incisors, 240 canines) in 120 patients were evaluated in this study. The ICC of this study was 0.885, which considered as good according to Cicchetti's criteria [23].

The result revealed that 84.58% (609/720) of anterior maxillary teeth had buccal bone of less than 1 mm. The SRP with probability of IIP are as follow: class I 87.91% (633/720) with 84.20% (533/633), class II 2.78% (20/720) with 70% (14/20), class III 0.42% (3/720) with 33.33% (1/3) and class IV 8.89% (64/720) with 0% respectively.

The distribution of SRP with probability of IIP are shown in Table 1.

The probability of IIP in all maxillary anterior teeth are 76.11% (548/720), with consist of central incisor 82.50% (198/240), lateral incisor 64.58% (155/240) and canine 81.25% (195/240). In this study, an individual tooth can have multiple implant positions. The probability of IIP for each relation (10 relations) is shown in Table 2.

The probability of IIP in screw-retained, incisal edge, angle screw and middle half are 2.64% (19/720), 26.11% (188/720), 55.42% (399/720) and 38.75% (279/720) respectively. The probability of IIP in each angulation is shown in Table 3.

The probability of IIP according to restoration type, cement-retained, straight screw-retained, angle screw-retained are 14.59% (80/548), 3.47% (19/548) and 85.40% (468/548) respectively. The probability of IIP in each restoration type is shown in Table 4.

**Table 1. Distribution of sagittal root position (SRP) with probability of IIP in the esthetic zone.**

| SRP | Class I | | | Class II | | | Class III | | | Class IV | | |
|---|---|---|---|---|---|---|---|---|---|---|---|---|
| Distribution | 633/720 (87.91%) | | | 20/720 (2.78%) | | | 3/720 (0.42%) | | | 64/720 (8.89%) | | |
| Probability | 533/633 (84.20%) | | | 14/20 (70%) | | | 1/3 (33.33%) | | | 0/64 (0%) | | |
| | Central incisor | Lateral incisor | Canine | Central incisor | Lateral incisor | Canine | Central incisor | Lateral incisor | Canine | Central incisor | Lateral incisor | Canine |
| Distribution | 213 (88.75%) | 183 (76.25%) | 237 (98.75%) | 9(3.75%) | 11(4.58%) | - | 1(0.42%) | 2(0.83%) | - | 17(7.08%) | 44(18.34%) | 3 (1.25%) |

**Table 2. Detailed probability of IIP by angulation groups and tooth type.**

| | Central incisor (240 teeth) | Lateral incisor (240 teeth) | Canine (240 teeth) | Total (720 teeth) |
|---|---|---|---|---|
| 1. Incisal edge (I) | 24(10%) | 29(12.08%) | 7(2.92%) | 60(8.33%) |
| 2. Screw-retained (S) | 1(0.42%) | - | - | 1(0.14%) |
| 3. Angle screw (A) | 57(23.75%) | 52(21.66%) | 21(8.75%) | 130(18.06%) |
| 4. Middle half (M) | 1(0.42%) | 6(2.5%) | 73(30.41%) | 80(11.11%) |
| 5. I, S | 8(3.33%) | - | - | 8(1.11%) |
| 6. I, A | 47(19.58%) | 10(4.17%) | 6(2.5%) | 63(8.75%) |
| 7. A, M | 26(10.83%) | 45(18.75%) | 78(32.5%) | 149(20.69%) |
| 8. I, S, A | 7(2.92%) | - | - | 7(0.97%) |
| 9. I, A, M | 24(10%) | 13(5.42%) | 10(4.17%) | 47(6.53%) |
| 10. I, S, A, M | 3(1.25%) | - | - | 3(0.42%) |
| **Total** | **198/240(82.5%)** | **155/240(64.58%)** | **195(81.25%)** | |

1. I = incisal edge, 2. S = screw, 3. A = angle, 4. M = middle, 5. I, S = incisal edge and screw, 6. I, A = incisal edge and angle, 7. A, M = angle and middle, 8. I, S, A = incisal edge, screw and angle, 9. I, A, M = incisal edge, angle and middle, 10. I, S, A, M = incisal edge, screw, angle and middle

Central incisor can position implant with diameter of 3.3 mm, 3.5 mm, 3.75 mm, 4.0 mm and 4.5 mm at 81.67% (196/240), 68.33% (164/240), 54.58% (131/240), 40.83% (98/240) and 5% (12/240) respectively and length of 10 mm, 12 mm, 14 mm, 16 mm and 18 mm at 22.5% (54/240), 62.92% (151/240), 80% (192/240), 73.33% (176/240) and 54.58% (131/240) respectively.

Lateral incisor can position implant with diameter of 2.9 mm, 3.3 mm, 3.5 mm, 3.75 mm and 4.0 mm at 57.92% (139/240), 51.25% (123/240), 36.25% (87/240), 26.67% (64/240) and 15% (36/240) respectively and length of 10 mm, 12 mm, 14 mm, 16 mm and 18 mm at 6.67% (16/240), 30% (72/240), 56.67% (136/240), 43.33% (104/240) and 30.83% (74/240) respectively.

Canine can position implant with diameter of 3.3 mm, 3.5 mm, 3.75 mm, 4.0 mm, 4.5 mm and 5.0 mm at 73.33% (176/240), 77.92% (187/240), 75.42% (181/240), 71.25% (171/240), 55% (132/240) and 23.75% (57/240) respectively and length of 10 mm, 12 mm, 14 mm, 16 mm and 18 mm at 7.5% (18/240), 25.42% (61/240), 49.58% (119/240), 58.75% (141/240) and 55% (132/240) respectively.

The distribution and the percentages of the diameter and length, which were used for IIP are shown in Table 5.

**Table 3. Summary of IIP probability by angulation groups and tooth type.**

| Angulation | Central incisor (240 teeth) | Lateral incisor (240 teeth) | Canine (240 teeth) | Total (720 teeth) |
|---|---|---|---|---|
| Incisal edge<br>1, 5, 6, 8, 9, 10 | 113(47.08%) | 52(21.67%) | 23(9.58%) | 188(26.11%) |
| Screw-retained<br>2, 5, 8, 10 | 19(7.92%) | 0 | 0 | 19(2.64%) |
| Angle screw<br>3, 6, 7, 8, 9, 10 | 164(68.33%) | 120(50%) | 115(47.92%) | 399(55.42%) |
| Middle half<br>4, 7, 9, 10 | 54(22.5%) | 64(26.67%) | 161(67.08%) | 279(38.75%) |

1. I = incisal edge, 2. S = screw, 3. A = angle, 4. M = middle, 5. I, S = incisal edge and screw, 6. I, A = incisal edge and angle, 7. A, M = angle and middle, 8. I, S, A = incisal edge, screw and angle, 9. I, A, M = incisal edge, angle and middle, 10. I, S, A, M = incisal edge, screw, angle and middle

**Table 4. Summary of restoration probability by restoration type and tooth with possible IIP.**

| Restoration type | Central incisor (198 teeth) | Lateral incisor (155 teeth) | Canine (195 teeth) | Total (548 teeth) |
|---|---|---|---|---|
| Cement-retained* | 1(0.51%) | 6(3.87%) | 73(37.43%) | 80(14.59%) |
| 4 | | | | |
| Straight screw-retained | 19(9.59%) | 0 | 0 | 19(3.47%) |
| 2, 5, 8, 10 | | | | |
| Angle screw-retained | 197(99.49%) | 149(96.13%) | 122(62.56%) | 468(85.40%) |
| 1, 2, 3, 5, 6, 7, 8, 9, 10 | | | | |

*Cement retained prosthesis is indicated when screw-retained prosthesis can't be achieved.

1. I = incisal edge, 2. S = screw, 3. A = angle, 4. M = middle, 5. I, S = incisal edge and screw, 6. I, A = incisal edge and angle, 7. A, M = angle and middle, 8. I, S, A = incisal edge, screw and angle, 9. I, A, M = incisal edge, angle and middle, 10. I, S, A, M = incisal edge, screw, angle and middle

**Table 5. Distributions of diameter and length of implant used in immediate implant placement in esthetic zone.**

| | Diameter | | | | Length | | |
|---|---|---|---|---|---|---|---|
| | Central incisor (240 teeth) | Lateral incisor (240 teeth) | Canine (240 teeth) | | Central incisor (240 teeth) | Lateral incisor (240 teeth) | Canine (240 teeth) |
| 2.9 mm | - | - | 139 | 57.92% | - | - | 6 mm | - | - | - | - | - | - |
| 3.3 mm | 196 | 81.67% | 123 | 51.25% | 176 | 73.33% | 8 mm | - | - | - | - | - | - |
| 3.5 mm | 164 | 68.33% | 87 | 36.25% | 187 | 77.92% | 10 mm | 54 | 22.5% | 16 | 6.67% | 18 | 7.5% |
| 3.75 mm | 131 | 54.58% | 64 | 26.67% | 181 | 75.42% | 12 mm | 151 | 62.92% | 72 | 30% | 61 | 25.42% |
| 4.0 mm | 98 | 40.83% | 36 | 15% | 171 | 71.25% | 14 mm | 192 | 80% | 136 | 56.67% | 119 | 49.58% |
| 4.5 mm | 12 | 5% | - | - | 132 | 55% | 16 mm | 176 | 73.33% | 104 | 43.33% | 141 | 58.75% |
| 5 mm | - | - | - | - | 57 | 23.75% | 18 mm | 131 | 54.58% | 74 | 30.83% | 132 | 55% |

## Discussion

The success criteria for immediate implant placement (IIP) in the esthetic zone including non-acute infection areas, presence of thick gingival biotype, engaging sufficient apical bone, with intact buccal bone [3]. Recent studies have shown that on average, the buccal bone in anterior maxillary teeth was less than 1 mm [24, 25]. This study found similar results with the majority of anterior maxillary teeth, 84.58% (609/720), having less than 1 mm of buccal bone thickness. Canine teeth 29.17% (70/240), were found most frequently to have a buccal bone thickness greater than 1 mm, followed by the lateral incisors and central incisors 11.25% (27/240) and 5.83% (14/240) respectively.

There was a 12.5% (90/720) probability of IIP in the group where the buccal bone is more than 1 mm, which leaves a probability of 63.61% (458/720) for the group of buccal bone that has less than 1 mm. In this group, there is a high risk for recession of the facial mucosa and significant flattening the soft tissue profile at the neck of implant prosthesis [1]. It is important to emphasize that the use of buccal bone augmentation [26], soft tissue graft [27], the combination of bone augmentation and soft tissue graft or alternative option such as partial tooth extraction therapy [28] can contribute to improve the overall esthetic result, and compensate with buccal contour remodeling in IIP.

Sagittal root position (SRP) could directly influence the position of IIP. In the study of Kan et al. [12], the author suggested that class I SRP was favorable and class IV SRP was regarded as a contraindication for IIP [12]. In this study, the distribution with probability results of class I and class IV SRP were in concordance with the study of Kan, the highest distribution and

probability is class I 87.42% (633/720) with probability of 84.20% (533/633) and class IV 8.89% (64/720) with the lowest probability of 0% (0/64). Surprisingly, the probability in class II is more than in class III, 70% (14/20) and 33.33% (1/3) respectively, which were different from previous study. These may due to sufficient apical bone to engage in traditional cement-retained implant prosthesis (incisal edge) in class II, whereas in class III inadequate apical bone causing implant to extended outward of alveolar housing.

Direction of implant prosthesis can influence the position of IIP as well. Cement-retained implant prosthesis has more advantages for esthetics, such as absence of screw access hole. In addition, long axis of implant coincides incisal edge, which creates a proper emergence angle that similar to the natural teeth approximately 15˚ [29]. While a screw-retained implant prosthesis is beneficial for retrievability, reducing complication from the left-over cement, which can further cause inflammation and bone loss [30, 31]. However, the disadvantages such as screw access hole might be esthetically compromised. Therefore, in the esthetic zone, the implant should be placed at cingulum area. However, the more palatal the implant placement, the more emergence angle and therefore makes a greater cervical contour [32]. The mentioned problem caused peri-implantitis and recession on labial surface [33]. Angulated screw channel avoids screw access hole to be placed in palatal area without compromised esthetic [34]. In addition, the position of implant that used of angulated screw channel is on labial aspect, which creates proper emergence angle. The results of this study revealed that if using traditional prosthetic screw with 0-degree compensation the prosthesis is limited to 14.59% (80/548) cement-retained with 3.47% (19/548) straight screw-retained prosthesis. By incorporating angulated screw with 25-degree compensation could increase the probability of screw-retained prosthesis to 85.40% (468/548). In individual tooth, an angulated screw channel could help increase the probability for screw-retained prosthesis in central incisor from 9.59% (19/198) to 99.49% (197/198), in lateral incisors from 0% to 96.13% (149/155), and in canine from 0% to 62.56% (122/195). Nevertheless, it's important to noted that an angle over 25˚ above limitation of angulated screw channel, IIP can still be performed but restricted to cement-retained prosthesis. In this study 14.59% (80/548) were restricted to only "middle half" angulation which result in cement-retained prosthesis. It is noteworthy that majority of cement-retained are with canine 91.25% (73/80) which coincide with majority of canine being SRP class I.

When considering in implant length, IIP required the engagement of 4 to 5 mm of bone apical to tooth socket [3], this result is increased implant length than the traditional implant placement. An implant length of 14 mm is the most frequently used in central incisor and lateral incisor 80% (192/240) and 56.67% (136/240) respectively. An implant length of 16 mm is the most frequently used for canine teeth 58.75% (141/240), as canine has the longest root length of 15.83 ± 1.49 mm [35]. Beyond the mentioned implant length could result in over extrusion of implant. When considering implant diameter, a diameter of 3.3 mm is the most frequently used for central incisors 81.67% (196/240). A diameter of 2.9 mm is the most frequently used in lateral incisor 57.92% (139/240), due to the least faciopalatal root dimension. For canine teeth with widest faciopalatal 6.67 mm [36], a diameter of 3.5 mm is the most frequently used 77.92% (187/240). Noteworthy in this study, the diameter of 2.9 mm is used in lateral incisor only, due to the manufacturer's recommendation. Previous study had shown that narrow diameter implant with < 3 mm has decrease survival rate [37] with increases the risk of implant fracture, abutment and screw failure [38, 39]. Even though, when simulating 2.9 mm diameter in central incisors and canines can increase the probability from 82.50% (198/240 teeth) to 87.50% (210/240) and 81.25% (195/240) to 82.08% (197/240).

In recent years, there are two very similar studies Kan et al. [19] and Erin et al. [40]. In Kan [19] study investigate the probability of using straight screw-channel screw-retained restoration for IIP in maxillary anterior teeth. The method was 2D implant planning using digital

template. The digital template for 3.5 mm diameter was used in central incisor and lateral incisor. The digital template for 4.3 mm diameter was used in central incisor and canine. The implant length was 13, 15 and 18 mm. In Erin [40] study was virtual implant planning in screw-retained implant direction using straight or angled screw channel abutment with different angulation (5˚, 10˚, 15˚). The process is not IIP and the implant diameter for each site was fix based on standards for each tooth type (canine and central incisor 4.3 mm, lateral incisor 3.5 mm). The implant length was 10 mm for all implant.

In this study was virtual implant planning to determine the probability of IIP in the esthetic zone. The additional implant angulations mentioned in more than two studies were cement-retained implant prosthesis, angulated screw channel at the 25˚ angulation and positioning of implant at the middle of individual tooth. Various size of diameter (2.9, 3.3, 3.5, 3.75, 4.0, 4.5, 5 mm) and length (6, 8, 10, 12, 14, 16, 18 mm) were included in this study to promote adaptability and suitability in IIP in the esthetic zone.

For limitation of this study to standardized the method, implant planning is limited to static predetermined angulation to anatomical relation and implant size. This could limit the probability as clinician can increase the depth of implant placement, modified implant angulation within restorable position. Future study is suggested to study the effect of probability associated with dynamic implant planning.

## Conclusion

Within the limitation of this CBCT study, thorough treatment planning and precise evaluation a probability of 76.11% (548/720) perform IIP. Straight screw-retained prosthesis allows for the lowest probability 3.47% (19/548) to perform IIP in the esthetic zone. Moreover, the use of angulated screw channel demonstrated to increase the probability of straight screw-retained prosthesis to 85.40% (468/548). However, the limit of angulated screw could not compensate for all IIP position with 14.59% (80/548) restricted to cement-retained prosthesis. A careful individual site planning is advised to maximize treatment outcome.

## Supporting information

**S1 Data.**
(XLSX)

## Author Contributions

**Conceptualization:** Pranai Nakaparksin.

**Data curation:** Watcharapon Radomsutthisarn.

**Formal analysis:** Watcharapon Radomsutthisarn.

**Investigation:** Watcharapon Radomsutthisarn.

**Methodology:** Kornkamol Kretapirom, Pranai Nakaparksin.

**Project administration:** Pranai Nakaparksin.

**Resources:** Watcharapon Radomsutthisarn, Kornkamol Kretapirom.

**Supervision:** Kornkamol Kretapirom, Pranai Nakaparksin.

**Visualization:** Watcharapon Radomsutthisarn, Kornkamol Kretapirom.

**Writing – original draft:** Watcharapon Radomsutthisarn.

**Writing – review & editing:** Kornkamol Kretapirom, Pranai Nakaparksin.

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
