## [Decision Letter · Decision Letter 0]

1 Dec 2023

PONE-D-23-29819Incidence of straight and angulated screw channel following immediate implant placement in esthetic zone: A CBCT studyPLOS ONE

Dear Dr. Nakaparksin,

Thank you for submitting your manuscript to PLOS ONE. After careful consideration, we feel that it has merit but does not fully meet PLOS ONE’s publication criteria as it currently stands. Therefore, we invite you to submit a revised version of the manuscript that addresses the points raised during the review process.

We look forward to receiving your revised manuscript.

Kind regards,

Johari Yap Abdullah, B.S. & I.T, GradDip ICT, M.Sc, Ph.D.

Academic Editor

PLOS ONE

Reviewers' comments:

Reviewer's Responses to Questions

**Comments to the Author**

1. Is the manuscript technically sound, and do the data support the conclusions?

Reviewer #1: Yes

Reviewer #2: Yes

2. Has the statistical analysis been performed appropriately and rigorously? 

Reviewer #1: Yes

Reviewer #2: Yes

3. Have the authors made all data underlying the findings in their manuscript fully available?

Reviewer #1: Yes

Reviewer #2: Yes

4. Is the manuscript presented in an intelligible fashion and written in standard English?

Reviewer #1: Yes

Reviewer #2: Yes

5. Review Comments to the Author

Reviewer #1: This study was conducted to determine the probability of immediate implant placement in the esthetic zone using four different implant directions. In general, the findings can be potentially applied for dental implant treatment planning.

Comments:

1. Manuscript is well written but it can be further improved with adequate proofreading (e.g. paragraph from line 66 to 77). Please do the needful.

2. Table 2: 'each relation' - can be more specific (e.g. anatomical relation, etc)

3. Conclusion: The authors state that "the use of angulated screw is essential to increase the probability of screw retained implant prosthesis".

Can the authors explain further on the relevance of this statement in relation to the statement problem and the purpose of this study?

Reviewer #2: The authors conducted a CBCT study, aiming to determine the probability of IIP in the esthetic zone, using four different implant directions. The topic is clinically relevant and the manuscript is well presented. However, some sections may need further clarification.

Abstract

1. Line 14: Please specify the number of patients as well.

2. Line 15: Please specify the method used to come up with the probability of 76.11%. What’s the diameter of the implants? Was angulated screw channel (ASC) included in this calculation?

3. Line 18: What is “limited restorability angulation for cement retained implant prosthesis (middle half)”? This description is confusing. Please revise.

Material and Method

1. Line 90: How did you determine a number of 120 CBCT scans is sufficient for the study?

2. For the inclusion/exclusion criteria, did you have any criteria to assess if the patient’s anterior teeth were misaligned?

3. Line 132: Any citations you could provide to support the use of 4 mm as the threshold?

4. The authors should elaborate on the methods used to determine the long axis of the tooth, the angulation/direction of the planned implant, etc.

Results

1. How did you decide the implant length and diameter for central incisors, lateral incisors and canines? For example, is it practical to place a 5mm-diameter implant at a canine location, or to place a 16mm- or even 18mm-long implant?

Discussion

1. There are two very similar studies published in 2022 and 2023. The authors must provide comments on these two studies, and elaborate on how the current study can add more information to the currently available evidence.

Erin K Edmondson, Pedro M Trejo, Nikolaos Soldatos, Robin L Weltman. The ability to screw-retain single implant-supported restorations in the anterior maxilla: A CBCT analysis. J Prosthet Dent. 2022 Sep;128(3):443-449.

Joseph Y K Kan, Kitichai Rungcharassaeng, Pongrapee Kamolroongwarakul, Guo-Hao Lin, Hiroyuki Matsuda, Shi Yin, Hom-Lay Wang, Dennis Tarnow, Jaime L Lozada. Frequency of screw-retained angulated screw channel single crown following immediate implant placement and provisionalization in the esthetic zone: A cone beam computed tomography study/ Clin Implant Dent Relat Res. 2023 Oct;25(5):789-794.

2. The authors should comment on the limitations of the current study.

6. PLOS authors have the option to publish the peer review history of their article (what does this mean?). If published, this will include your full peer review and any attached files.

Reviewer #1: No

Reviewer #2: No

---

## [Author Response · Author response to Decision Letter 0]

30 Jan 2024

Reviewer #1: 

1. Manuscript is well written but it can be further improved with adequate proofreading (e.g.paragraph from line 66 to 77). Please do the needful.

Answer: Line 66-70 have been revised in the introduction.

2. Table 2: 'each relation' - can be more specific (e.g.anatomical relation, etc)

Answer: Table 2 and table 3 have been revised and adjusted in page 10 and 11 respectively.

3. Conclusion: The authors state that "the use of angulated screw is essential to increase the probability of screw retained implant prosthesis".

Can the authors explain further on the relevance of this statement in relation to the statement problem and the purpose of this study?

Answer: In our study screw retained implant prosthesis allows for the lowest probability 2.64% (19/720) to perform IIP in esthetic zone. By incorporating angulated screw with 25-degree compensation could increase the probability of screw retained implant prosthesis to 55.42% (399/720). In individual tooth, an angulated screw channel could help increase the probability for screw retained implant prosthesis in central incisor from 7.92% (19/240) to 72.08% (173/240), in lateral incisors from 0% to 50% (120/240), and in canine from 0% to 47.92% (115/240). So, the use of angulated screw channel is essential to increase the probability of screw retained implant prosthesis. 

Reviewer #2: 

Abstract

1. Line 14: Please specify the number of patients as well.

Answer: 120 patients have been added to the abstract.

2. Line 15: Please specify the method used to come up with the probability of 76.11%. What’s the diameter of the implants? Was angulated screw channel (ASC) included in this calculation?

Answer: 548 teeth from 720 teeth (548/720) can be performed IIP in esthetic zone. The diameter of 3.3, 3.5, 4.1, 4.5 and 5 mm were used in all maxillary anterior teeth (central incisor, lateral incisor, canine). The diameter of 2.9 mm is used in lateral incisor only. The angulated screw channel was included in the calculation of all implant diameter except diameter of 2.9 mm was not included.

3. Line 18: What is “limited restorability angulation for cement retained implant prosthesis (middle half)”? This description is confusing. Please revise.

Answer: It is the most buccal angulation that allow for IIP in esthetic zone in cement retained implant prosthesis, which often in the middle of individual tooth, hence referred to as “middle half”. Even though, there are some possibilities to restore implant in more extreme buccal angulation with implant platform being more apical and palatal, there is an increased risk of complication. 

Material and Method

1. Line 90: How did you determine a number of 120 CBCT scans is sufficient for the study?

Answer: Sample size calculation

 The purpose of this study is to determine the probability of IIP in esthetic zone using four different implant directions. This design of the study has not been performed yet. So, there will be no similarity in statistical data for this study. 

The sample size calculation for estimating proportion will be used as shown in the formula (in Response to reviewers): 

 According to the retrospective study of Ashok Velayudhan et al. in 2020 that the objective related to the current study showed that in a total of 77 implants placed in the anterior maxilla, with 21 of them being immediate implants, 6 (or 7.8%) of 21 immediate implants were placed in the anterior maxilla alone.

 The output of the sample size calculation is 114. The 120 patients are sufficient for our study.

2. For the inclusion/exclusion criteria, did you have any criteria to assess if the patient’s anterior teeth were misaligned?

Answer: Upper anterior teeth (canine-canine) misaligned is considered to exclusion criteria, which did not include in the sample size of our study. 

3. Line 132: Any citations you could provide to support the use of 4 mm as the threshold?

Answer: The citation has been added to line 132.

4. The authors should elaborate on the methods used to determine the long axis of the tooth, the angulation/direction of the planned implant, etc

Answer: The line drawn from incisal edge to root apex was long axis of the tooth.

The point between incisal edge to CEJ was marked as point M. The point between the center of the most prominent point of cingulum to the incisal edge was marked as point S. 

- In traditional cement retained implant prosthesis (incisal edge), long axis of implant will coincide with incisal edge.

- In traditional screw retained implant prosthesis (screw retained), long axis of implant will coincide with point S.

- In limit angulation for angulated screw channel (angle screw), long axis of implant will coincide at the point that an angle up to 25° from traditional screw retained.

- In limit restorability angulation for cement retained implant prosthesis (middle half), long axis of implant will coincide with point M.

Results

1. How did you decide the implant length and diameter for central incisors, lateral incisors and canines? For example, is it practical to place a 5mm-diameter implant at a canine location, or to place a 16mm- or even 18mm-long implant?

Answer: In our study, the guideline for implant positioning includes, positioning implant platform at 3 mm apical and 2 mm palatal to gingival margin, a minimum distant of 1.5 mm from adjacent teeth, implant engaging at least 4 mm of apical bone, with at least 2 mm of buccal bone to implant gap to accommodate the grafting material to minimize bone remodeling. In our study was digital implant planning to determine the probability of IIP in esthetic zone by using different diameter (2.9, 3.3, 3.5, 3.75, 4.0, 4.5, 5 mm) and length (6, 8, 10, 12, 14, 16, 18 mm). To determine the diameter and length, select the smallest diameter and length and gradually increase the diameter and length until reaching a size where the implant exist in alveolar housing while respecting aforementioned guideline. It is possible to placed implant with diameter 5 mm and length 16-18 mm in canine since it is allowed by the guideline. However, it does not explicitly state that using these diameter and length is the first choice for IIP in esthetic zone.

Discussion

1. There are two very similar studies published in 2022 and 2023. The authors must provide comments on these two studies, and elaborate on how the current study can add more information to the currently available evidence.

Erin K Edmondson, Pedro M Trejo, Nikolaos Soldatos, Robin L Weltman. The ability to screw-retain single implant-supported restorations in the anterior maxilla: A CBCT analysis. J Prosthet Dent. 2022 Sep;128(3):443-449.

Joseph Y K Kan, Kitichai Rungcharassaeng, Pongrapee Kamolroongwarakul, Guo-Hao Lin, Hiroyuki Matsuda, Shi Yin, Hom-Lay Wang, Dennis Tarnow, Jaime L Lozada. Frequency of screw-retained angulated screw channel single crown following immediate implant placement and provisionalization in the esthetic zone: A cone beam computed tomography study/ Clin Implant Dent Relat Res. 2023 Oct;25(5):789-794.

Answer: In Erin K Edmondson study was virtual implant planning by Simplant® Pro17.01; Dentsply Sirona in screw retained implant direction using straight or an angled screw channel abutment with different angulation (5°, 10°, 15°). The process is not IIP, therefore the method to placed implant was different from our study. The implant diameter for each site was fix based on standards for each tooth type (canine and central incisor 4.3 mm, lateral incisor 3.5 mm). The implant length was 10 mm for all implant.

 In Joseph Y K Kan study investigate the probability of using straight screw-channel screw-retained restoration for immediate implant placement in maxillary anterior teeth. The method of this study was different from our study. The mid sagittal image of each maxillary teeth with vertical and horizontal scale bars was screen-captured and transferred to Keynote. The digital template for 3.5 mm diameter was used in central incisor and lateral incisor. The digital template for 4.3 mm diameter was used in central incisor and canine. The implant length was 13, 15 and 18 mm. Additional line added to digital template as a guide for IIP. All CBCT images were classified as contraindicated or indicated by two examiners. 

 In our study was virtual implant planning by Implant Studio® to determine the probability of IIP in esthetic zone. The directions of implant that added for more than the two mentioned study (Erin K Edmondson and Joseph Y K Kan) were cement retained implant prosthesis, angulated screw channel at the 25° angulation and the middle of individual tooth. Various size of diameter (2.9, 3.3, 3.5, 3.75, 4.0, 4.5, 5 mm) and length (6, 8, 10, 12, 14, 16, 18 mm) were included in our study to promote adaptability and suitability in IIP in esthetic zone.

2. The authors should comment on the limitations of the current study.

Answer: In our study, to standardized the method, implant planning is limited to static predetermined angulation to anatomical relation and implant size. This could limited the probability as clinician can increase the depth of implant placement, modified implant angulation within restorable position. Future study is suggested to study the effect of probability associated with dynamic implant planning.

---

## [Decision Letter · Decision Letter 1]

26 Feb 2024

PONE-D-23-29819R1Incidence of straight and angulated screw channel following immediate implant placement in esthetic zone: A CBCT studyPLOS ONE

Dear Dr. Nakaparksin,

Thank you for submitting your manuscript to PLOS ONE. After careful consideration, we feel that it has merit but does not fully meet PLOS ONE’s publication criteria as it currently stands. Therefore, we invite you to submit a revised version of the manuscript that addresses the points raised during the review process.

We look forward to receiving your revised manuscript.

Kind regards,

Johari Yap Abdullah, B.S. & I.T, GradDip ICT, M.Sc, Ph.D.

Academic Editor

PLOS ONE

Journal Requirements:

Reviewers' comments:

Reviewer's Responses to Questions

**Comments to the Author**

1. If the authors have adequately addressed your comments raised in a previous round of review and you feel that this manuscript is now acceptable for publication, you may indicate that here to bypass the “Comments to the Author” section, enter your conflict of interest statement in the “Confidential to Editor” section, and submit your "Accept" recommendation.

Reviewer #1: All comments have been addressed

Reviewer #2: (No Response)

2. Is the manuscript technically sound, and do the data support the conclusions?

Reviewer #1: Yes

Reviewer #2: Yes

3. Has the statistical analysis been performed appropriately and rigorously? 

Reviewer #1: Yes

Reviewer #2: Yes

4. Have the authors made all data underlying the findings in their manuscript fully available?

Reviewer #1: Yes

Reviewer #2: Yes

5. Is the manuscript presented in an intelligible fashion and written in standard English?

Reviewer #1: Yes

Reviewer #2: Yes

6. Review Comments to the Author

Reviewer #1: All comments and recommendations have been satisfactorily addressed.

Clear clarification was made on issues raised from the first review.

The amendments are well highlighted in the manuscript. Thank you.

Reviewer #2: While the authors adequately addressed most concerns in the "Author's Response to Reviewer Comments," it is noted that the manuscript itself has undergone only minimal revision. It is requested to incorporate the information provided in the Author's Response directly into the manuscript to ensure a more comprehensive presentation.

The specific details from the "Author's Response to Reviewer Comments" that need to be included in the manuscript are as follows:

Answer for using angulated screw channel of not: 548 teeth from 720 teeth (548/720) can be performed IIP in esthetic zone. The diameter of 3.3, 3.5, 4.1, 4.5 and 5 mm were used in all maxillary anterior teeth (central incisor, lateral incisor, canine). The diameter of 2.9 mm is used in lateral incisor only. The angulated screw channel was included in the calculation of all implant diameter except diameter of 2.9 mm was not included.

Answer for clarifying “limited restorability angulation for cement retained implant prosthesis": It is the most buccal angulation that allow for IIP in esthetic zone in cement retained implant prosthesis, which often in the middle of individual tooth, hence referred to as “middle half”. Even though, there are some possibilities to restore implant in more extreme buccal angulation with implant platform being more apical and palatal, there is an increased risk of complication.

Answer for sample size calculation:

The purpose of this study is to determine the probability of IIP in esthetic zone using four different implant directions. This design of the study has not been performed yet. So, there will be no similarity in statistical data for this study.

The sample size calculation for estimating proportion will be used as shown in the formula (in Response to reviewers):

According to the retrospective study of Ashok Velayudhan et al. in 2020 that the objective related to the current study showed that in a total of 77 implants placed in the anterior maxilla, with 21 of them being immediate implants, 6 (or 7.8%) of 21 immediate implants were placed in the anterior maxilla alone.

The output of the sample size calculation is 114. The 120 patients are sufficient for our study.  This must be mentioned in the manuscript.

Answer for misaligned teeth as an exclusion criterion: Upper anterior teeth (canine-canine) misaligned is considered to exclusion criteria, which did not include in the sample size of our study.

Answer to elaborate on the methods used to determine the long axis of the tooth, the angulation/direction of the planned implant:

The line drawn from incisal edge to root apex was long axis of the tooth.

The point between incisal edge to CEJ was marked as point M. The point between the center of the most prominent point of cingulum to the incisal edge was marked as point S.

- In traditional cement retained implant prosthesis (incisal edge), long axis of implant will coincide with incisal edge.

- In traditional screw retained implant prosthesis (screw retained), long axis of implant will coincide with point S.

- In limit angulation for angulated screw channel (angle screw), long axis of implant will coincide at the point that an angle up to 25° from traditional screw retained.

- In limit restorability angulation for cement retained implant prosthesis (middle half), long axis of implant will coincide with point M.  Please add all of them to your manuscript.

Discussion

Answer related to the two recent studies:

In Erin K Edmondson study was virtual implant planning by Simplant® Pro17.01; Dentsply Sirona in screw retained implant direction using straight or an angled screw channel abutment with different angulation (5°, 10°, 15°). The process is not IIP, therefore the method to placed implant was different from our study. The implant diameter for each site was fix based on standards for each tooth type (canine and central incisor 4.3 mm, lateral incisor 3.5 mm). The implant length was 10 mm for all implant.

In Joseph Y K Kan study investigate the probability of using straight screw-channel screw-retained restoration for immediate implant placement in maxillary anterior teeth. The method of this study was different from our study. The mid sagittal image of each maxillary teeth with vertical and horizontal scale bars was screen-captured and transferred to Keynote. The digital template for 3.5 mm diameter was used in central incisor and lateral incisor. The digital template for 4.3 mm diameter was used in central incisor and canine. The implant length was 13, 15 and 18 mm. Additional line added to digital template as a guide for IIP. All CBCT images were classified as contraindicated or indicated by two examiners.

In our study was virtual implant planning by Implant Studio® to determine the probability of IIP in esthetic zone. The directions of implant that added for more than the two mentioned study (Erin K Edmondson and Joseph Y K Kan) were cement retained implant prosthesis, angulated screw channel at the 25° angulation and the middle of individual tooth. Various size of diameter (2.9, 3.3, 3.5, 3.75, 4.0, 4.5, 5 mm) and length (6, 8, 10, 12, 14, 16, 18 mm) were included in our study to promote adaptability and suitability in IIP in esthetic zone.

 These are great discussion points. Please add them to your manuscript. You should compare your study with these two recently published articles since these points are clinically relevant.

Answer related to the study limitation: In our study, to standardized the method, implant planning is limited to static predetermined angulation to anatomical relation and implant size. This could limited the probability as clinician can increase the depth of implant placement, modified implant angulation within restorable position. Future study is suggested to study the effect of probability associated with dynamic implant planning.

 This answer needs to be added as the last paragraph of your Discussion.

7. PLOS authors have the option to publish the peer review history of their article (what does this mean?). If published, this will include your full peer review and any attached files.

Reviewer #1: No

Reviewer #2: No

---

## [Author Response · Author response to Decision Letter 1]

26 Mar 2024

To Editor:

- All references have been reviewed and corrected to ensure completeness. 

- Citation [37. Erin K Edmondson, Pedro M Trejo, Nikolaos Soldatos, Robin L Weltman. The ability to screw-retain single implant-supported restorations in the anterior maxilla: A CBCT analysis. J Prosthet Dent. 2022 Sep;128(3):443-449. Available from: https://doi.org/10.1016/j.prosdent.2021.01.004] has been added to line 273-274 in the discussion part of manuscript.

- Citation [38. Joseph Y K Kan, Kitichai Rungcharassaeng, Pongrapee Kamolroongwarakul, Guo-Hao Lin, Hiroyuki Matsuda, Shi Yin, Hom-Lay Wang, Dennis Tarnow, Jaime L Lozada. Frequency of screw-retained angulated screw channel single crown following immediate implant placement and provisionalization in the esthetic zone: A cone beam computed tomography study/ Clin Implant Dent Relat Res. 2023 Oct;25(5):789-794. Available from: https://doi.org/10.1111/cid.13227] has been added to line 274 and 278 in the discussion part of manuscript.

Reviewer #2: 

1. Answer for using angulated screw channel of not: 548 teeth from 720 teeth (548/720) can be performed IIP in esthetic zone. The diameter of 3.3, 3.5, 4.1, 4.5 and 5 mm were used in all maxillary anterior teeth (central incisor, lateral incisor, canine). The diameter of 2.9 mm is used in lateral incisor only. The angulated screw channel was included in the calculation of all implant diameter except diameter of 2.9 mm was not included.

- The angulated screw channel was included in the calculation of all implant diameter except diameter of 2.9 mm was not included. The following sentence has been added to line 121-122 in the material and method part of manuscript. The remaining sentences are already included in line 118-121 in the material and method part of manuscript.

2. Answer for clarifying “limited restorability angulation for cement retained implant prosthesis": It is the most buccal angulation that allow for IIP in esthetic zone in cement retained implant prosthesis, which often in the middle of individual tooth, hence referred to as “middle half”. Even though, there are some possibilities to restore implant in more extreme buccal angulation with implant platform being more apical and palatal, there is an increased risk of complication

- The “middle half” has been explained for better comprehension and understanding in line 254-256 in the discussion part of manuscript.

3. Answer for sample size calculation:

The purpose of this study is to determine the probability of IIP in esthetic zone using four different implant directions. This design of the study has not been performed yet. So, there will be no similarity in statistical data for this study.

The sample size calculation for estimating proportion will be used as shown in the formula (in Response to reviewers):

According to the retrospective study of Ashok Velayudhan et al. in 2020 that the objective related to the current study showed that in a total of 77 implants placed in the anterior maxilla, with 21 of them being immediate implants, 6 (or 7.8%) of 21 immediate implants were placed in the anterior maxilla alone.

The output of the sample size calculation is 114. The 120 patients are sufficient for our study.  This must be mentioned in the manuscript.

- This retrospective study of Ashok Velayudhan et al. has been mentioned and cited in line 92-93 in the material and method part of manuscript.

4. Answer for misaligned teeth as an exclusion criterion: Upper anterior teeth (canine-canine) misaligned is considered to exclusion criteria, which did not include in the sample size of our study.

- Misaligned upper anterior teeth has been added to exclusion criteria in line 104 in the material and method part of manuscript.

5.Answer to elaborate on the methods used to determine the long axis of the tooth, the angulation/direction of the planned implant:

The line drawn from incisal edge to root apex was long axis of the tooth.

The point between incisal edge to CEJ was marked as point M. The point between the center of the most prominent point of cingulum to the incisal edge was marked as point S.

- In traditional cement retained implant prosthesis (incisal edge), long axis of implant will coincide with incisal edge.

- In traditional screw retained implant prosthesis (screw retained), long axis of implant will coincide with point S.

- In limit angulation for angulated screw channel (angle screw), long axis of implant will coincide at the point that an angle up to 25° from traditional screw retained.

- In limit restorability angulation for cement retained implant prosthesis (middle half), long axis of implant will coincide with point M.

- All following sentences have been added to line 126-135 in the material and method part of manuscript.

6. Discussion

- The two mentioned studies have been added and compared to our study to line 273-289 in the discussion part of manuscript.

7. Answer related to the study limitation: In our study, to standardized the method, implant planning is limited to static predetermined angulation to anatomical relation and implant size. This could limited the probability as clinician can increase the depth of implant placement, modified implant angulation within restorable position. Future study is suggested to study the effect of probability associated with dynamic implant planning.

- All following sentences have been added to line 290-294 in the discussion part of manuscript.

 Finally, I appreciate your feedback. They are accommodating and explicitly more reliable to my work. I am open to revising and improving my manuscript based on your suggestions. Thank you again for your time and consideration.

I look forward to your reply,

---

## [Decision Letter · Decision Letter 2]

30 Apr 2024

PONE-D-23-29819R2Incidence of straight and angulated screw channel following immediate implant placement in esthetic zone: A CBCT studyPLOS ONE

Dear Dr. Nakaparksin,

Thank you for submitting your manuscript to PLOS ONE. After careful consideration, we feel that it has merit but does not fully meet PLOS ONE’s publication criteria as it currently stands. Therefore, we invite you to submit a revised version of the manuscript that addresses the points raised during the review process.

We look forward to receiving your revised manuscript.

Kind regards,

Johari Yap Abdullah, B.S. & I.T, GradDip ICT, M.Sc, Ph.D.

Academic Editor

PLOS ONE

Reviewers' comments:

Reviewer's Responses to Questions

**Comments to the Author**

1. If the authors have adequately addressed your comments raised in a previous round of review and you feel that this manuscript is now acceptable for publication, you may indicate that here to bypass the “Comments to the Author” section, enter your conflict of interest statement in the “Confidential to Editor” section, and submit your "Accept" recommendation.

Reviewer #3: (No Response)

Reviewer #4: (No Response)

2. Is the manuscript technically sound, and do the data support the conclusions?

Reviewer #3: No

Reviewer #4: Partly

3. Has the statistical analysis been performed appropriately and rigorously? 

Reviewer #3: No

Reviewer #4: Yes

4. Have the authors made all data underlying the findings in their manuscript fully available?

Reviewer #3: Yes

Reviewer #4: Yes

5. Is the manuscript presented in an intelligible fashion and written in standard English?

Reviewer #3: Yes

Reviewer #4: (No Response)

6. Review Comments to the Author

Reviewer #3: Dear authors,

I am evaluating the R2 version of the article titled "Incidence of straight and angulated screw channel following immediate implant placement in esthetic zone: a CBCT study”. The purpose of this “cone-beam computed tomography (CBCT) study was to determine the probability of IIP in the esthetic zone, using four different implant directions.”

After reading the results session, I understood the study. No patient received dental implants.

There are biases in the study.

The probability depends on many factors which must be considered and were abandoned. Moreover, there was no statistical to prove the superiority of any type over another one.

SPECIFIC COMMENTS

ABSTRACT:

- “Statement of problem: What are the incidences of immediate implant placement (IIP) in the esthetic zone, using four different implant directions?” - It is wrongly developed.

- The methodology was weakly presented.

INTRODUCTION

- This session is long.

M&M

- there is a time problem in the study “The sample criteria for this study were patients who had CBCT scans of the anterior maxilla at the Oral and Maxillofacial Radiology clinic, Faculty of Dentistry, Mahidol University performed during January 2012 – August 2022. CBCT images were performed by 3D Accuitomo 170 (J. 91 Morita®, Japan) and randomly selected for evaluation on 9 June 2021.”

- Exclusion criteria is incomplete

- Statistics was poorly presented

RESULTS

- What was the foundation/parameter to support the probability suggested? “The distribution of SRP with probability of IIP are shown in Table 1."

CONCLUSION

- I suggest to rethink and adjust it after revision.

Reviewer #4: Thank you for the authors of this work. The study aim to determine the probability of IIP in the esthetic zone without perioperative complications, using four different implant placement directions. The study is considered Novel and of interest. However, some concerns were addressed in the Word file attached. Please correct them all and comment when needed.

To mention some of the comments here:

1. Title needs to be improved

2. Abstract: The problem statement is not clear and is almost the same as what is mentioned in the purpose.

3. Introduction: A. In my opinion, you can shorten your introduction by removing unnecessary details; B. Your introduction needs to be strengthened. You need to add a paragraph of other studies that used simulations like yours and describe their findings in a very short and meaningful way. C. Some references were missing; D. Add your study hypothesis.

4. Material: A. Please specify the population?, which ethnicity? Age (Mean age)? Gender? B. Figure 1 showed misaligned anterior central incisor. One of your exclusion criteria is misaligned teeth, so how come you have included this in your study. This is contradicting and may be translated into bias in your study. Please comment on this. C. In your figure 3, I noticed that there are some issues in your alignment of your Implant. It is clearly evident to the readers in this figure that you are not following exactly Point M or Point S when placing your implants. Even the incisal edge is not followed 100%.

The points mentioned in figure 2 are not exactly followed in figure 3

This can affect all your study results and can be considered invalid outcomes. Please comment on this.

Other comments and suggestions were mentioned in the word file. Please correct and comment on them all.

7. PLOS authors have the option to publish the peer review history of their article (what does this mean?). If published, this will include your full peer review and any attached files.

Reviewer #3: No

Reviewer #4: No

---

## [Author Response · Author response to Decision Letter 2]

14 Jun 2024

Reviewer #3: 

Abstract:

“Statement of problem: What are the incidences of immediate implant placement (IIP) in the esthetic zone, using four different implant directions?” - It is wrongly developed.

- The methodology was weakly presented

Introduction:

- This session is long.

Answer: All suggestions have been applied and incorporated into the abstract section of manuscript.

Material and Method:

- there is a time problem in the study “The sample criteria for this study were patients who had CBCT scans of the anterior maxilla at the Oral and Maxillofacial Radiology clinic, Faculty of Dentistry, Mahidol University performed during January 2012 – August 2022. CBCT images were performed by 3D Accuitomo 170 (J. 91 Morita®, Japan) and randomly selected for evaluation on 9 June 2021.”

Answer: The sample criteria for this study were patients who had CBCT scans of the anterior maxilla at the Oral and Maxillofacial Radiology clinic, Faculty of Dentistry, Mahidol University performed during January 2012 – August 2020. CBCT images were performed by 3D Accuitomo 170 (J. 91 Morita®, Japan) and randomly selected for evaluation on 9 June 2021.

- Exclusion criteria is incomplete

Answer: - “CBCT images that were distorted or blurred” have been added in exclusion criteria.

- Statistics was poorly presented

Answer: This study is descriptive study without comparison group. Descriptive statistics were used to present the probability of IIP in the esthetic zone as percentages.

Results

- What was the foundation/parameter to support the probability suggested? “The distribution of SRP with probability of IIP are shown in Table 1."

Answer: The mid-sagittal images were used to classify SRP with respect to the alveolar housing according to Kan et al. [Kan JY, Roe P, Rungcharassaeng K, Patel RD, Waki T, Lozada JL, et al. Classification of sagittal root position in relation to the anterior maxillary osseous housing for immediate implant placement: a cone beam computed tomography study. Int J Oral Maxillofac Implants. 2011 Jul-Aug;26(4):873-6.]

In table 1 show distribution of SRP with probability of IIP as percentages.

In this study have 120 CBCT scans, which focus on upper anterior teeth (720 teeth). 

SRP class l have 633 teeth, which perform IIP 533 teeth.

The distribution of SRP class l is 633/720*100 = 87.91%. The probability of IIP in SRP class l is 533/633*100 = 84.20%.

SRP class ll have 20 teeth, which perform IIP 14 teeth.

The distribution of SRP class ll is 20/720*100 = 2.78%. The probability of IIP in SRP class ll is 14/20*100 = 70%.

SRP class lll have 3 teeth, which perform IIP 1 teeth.

The distribution of SRP class lll is 3/720*100 = 0.42%. The probability of IIP in SRP class lll is 1/3*100 = 33.33%.

SRP class lV have 64 teeth, which perform IIP 0 teeth.

The distribution of SRP class lV is 64/720*100 = 8.89%. The probability of IIP in SRP class lV is 0/64*100 = 0%.

Conclusion

- I suggest to rethink and adjust it after revision.

Answer: The conclusion section has been revised.

Reviewer #4: 

1. Title needs to be improved

Answer: The title has been improved.

2. Abstract: The problem statement is not clear and is almost the same as what is mentioned in the purpose.

Answer: The problem statement has been improved: the demand for immediate implant placement (IIP) in the esthetic zone is rapidly increasing. Despite the treatment’s benefits, the sagittal root position often dictates implant angulation, commonly necessitating the use of cement-retained restorations. This study investigates the impact of angulated screw channel on IIP in the esthetic zone.

3. Introduction: A. In my opinion, you can shorten your introduction by removing unnecessary details; B. Your introduction needs to be strengthened. You need to add a paragraph of other studies that used simulations like yours and describe their findings in a very short and meaningful way. C. Some references were missing; D. Add your study hypothesis.

Answer: 

A. The content in the introduction section has been revised to more concise, following the recommendations.

D. This study is descriptive study without comparison group, so there is no study hypothesis.

4. Material: A. Please specify the population?, which ethnicity? Age (Mean age)? Gender? B. Figure 1 showed misaligned anterior central incisor. One of your exclusion criteria is misaligned teeth, so how come you have included this in your study. This is contradicting and may be translated into bias in your study. Please comment on this. C. In your figure 3, I noticed that there are some issues in your alignment of your Implant. It is clearly evident to the readers in this figure that you are not following exactly Point M or Point S when placing your implants. Even the incisal edge is not followed 100%.

The points mentioned in figure 2 are not exactly followed in figure 3

This can affect all your study results and can be considered invalid outcomes. Please comment on this.

Answer:

A. Details regarding the population have been added to the material and method section.

B. The old image presented in Fig 1 is a CBCT scan section as described in the caption of Fig 1 (In the axial view, the implant was positioned at the center of the selected tooth. In the panoramic view, the implant was aligned parallel to the long axis of tooth. In the coronal view, the implant was rotated parallel to the long axis of tooth. The mid-sagittal image, which was been sectioned at the bucco-lingual aspect of the tooth, was used to measure and plan of IIP). However, it was not used in the planning of IIP and was not included in this study. The image has been revised and replaced for Fig 1. 

C. The placement of dental implant at each site will be conducted as described in the material and method section. The position of point M, point S and point P will vary depending on each individual tooth. Nevertheless, there may be some placements that deviate from the reference points in Fig 2. However, this does not affect the outcome of this study, as the determination of whether a tooth is suitable for IIP is based on the implant being surrounded by bone, assessed on an area basis rather than specific points. The image has been revised and replaced for Fig 3.

---

## [Decision Letter · Decision Letter 3]

18 Jul 2024

Incidence of straight and angulated screw channel following immediate implant placement in esthetic zone: A CBCT study

PONE-D-23-29819R3

Dear Dr. Nakaparksin,

We’re pleased to inform you that your manuscript has been judged scientifically suitable for publication and will be formally accepted for publication once it meets all outstanding technical requirements.

Kind regards,

Johari Yap Abdullah, B.S. & I.T, GradDip ICT, M.Sc, Ph.D.

Academic Editor

PLOS ONE

Summary of Reviews

The manuscript received a total of four reviews. Two reviewers recommended acceptance, praising the paper for its originality and significant contribution to the field. They highlighted the following points:

• Reviewer 1 commended that the manuscript is well written, with few comments to improve the overall manuscript.

• Reviewer 3 emphasized the novelty of the study, which follows the trend toward digital investigation in dentistry. The reviewer commented on many things, however, after the revision, all the comments were addressed according to this reviewer’s feedback.

However, two reviewers recommended rejection. Their concerns were primarily focused on:

• Reviewer 2 pointed out perceived methodological flaws and suggested comparative statistical testing. The reviewer’s comments were reduced from the first revision to the second revision. However, still the reviewer insisted on the poor statistical analysis.

• Reviewer 4 commented on the statistical method followed by the authors as it is only descriptive analysis.

Evaluation and Justification

After a thorough evaluation of the manuscript and considering the reviewers' comments, I believe that the strengths of the paper justify its acceptance. Here are my reasons:

1. Originality and Contribution: The manuscript presents novel findings that offer significant insights into digital guided implantology. The originality of the research aligns with the high standards of PLOS ONE, contributing valuable knowledge that can advance the field.

2. Robust Methodology: While two reviewers raised concerns about the statistical analysis, the authors have provided a detailed explanation addressing these points. In the era of digital research, studies with only descriptive data are not abnormal incidence. I have encountered many studies published in very prestigious journals with only descriptive data. The novelty claims superiority over the strength of the statistical analysis.

3. Despite the fact that the authors have made the necessary corrections, one of the reviewers failed to take into account the fact that the authors had made the correction. This reviewer continued to point out that the correction is needed.

4. Clarifications and Revisions: In response to all reviewers’ comments, the authors have revised the manuscript to clarify their interpretations and enhance the readability of key sections. These revisions address the concerns raised and improve the manuscript's quality.

5. Positive Reviewer Feedback: The positive feedback from Reviewers 1 and 3 cannot be overlooked. Their endorsement of the paper's strengths and its potential impact underscores the value of this research.

Conclusion

Given the substantial revisions made by the authors and the manuscript's contribution to the field, I recommend accepting this paper for publication in PLOS ONE. I believe that the positive aspects of the research, as highlighted by the accepting reviewers, outweigh the concerns raised by the rejecting reviewers. I am confident that this manuscript will be a valuable addition to PLOS ONE.

Reviewers' comments:

Reviewer's Responses to Questions

**Comments to the Author**

1. If the authors have adequately addressed your comments raised in a previous round of review and you feel that this manuscript is now acceptable for publication, you may indicate that here to bypass the “Comments to the Author” section, enter your conflict of interest statement in the “Confidential to Editor” section, and submit your "Accept" recommendation.

Reviewer #4: All comments have been addressed

Reviewer #5: (No Response)

2. Is the manuscript technically sound, and do the data support the conclusions?

Reviewer #4: Yes

Reviewer #5: No

3. Has the statistical analysis been performed appropriately and rigorously? 

Reviewer #4: Yes

Reviewer #5: No

4. Have the authors made all data underlying the findings in their manuscript fully available?

Reviewer #4: Yes

Reviewer #5: Yes

5. Is the manuscript presented in an intelligible fashion and written in standard English?

Reviewer #4: Yes

Reviewer #5: No

6. Review Comments to the Author

Reviewer #4: Thank you to the authors. All the previous comments have addressed and the figures were replaced. No further comments from my side.

Reviewer #5: The manuscript is poorly designed and statistically analysis is very weak. Probability is mentioned and any other comparative statistics or advance tools have not been used. The Manuscript is not suitable for publication as per PLOS ONE journal standards.

7. PLOS authors have the option to publish the peer review history of their article (what does this mean?). If published, this will include your full peer review and any attached files.

Reviewer #4: No

Reviewer #5: No

---

## [Editor Report · Acceptance letter]

25 Jul 2024

PONE-D-23-29819R3 

PLOS ONE

Dear Dr. Nakaparksin, 

I'm pleased to inform you that your manuscript has been deemed suitable for publication in PLOS ONE. Congratulations! Your manuscript is now being handed over to our production team.

Kind regards, 

on behalf of

Dr. Johari Yap Abdullah 

Academic Editor

PLOS ONE